biomaterials/bioengineering/cellular biology

mechanobiology, cornea, substrate stiffness, epithelium

**Author for correspondence:**
Mark Ahearne
e-mail: ahearnm@tcd.ie

# Influence of polydimethylsiloxane substrate stiffness on corneal epithelial cells

## Sophia Masterton[1,2] and Mark Ahearne[1,2]

[1]Department of Mechanical and Manufacturing Engineering, School of Engineering, and [2]Trinity Centre for Biomedical Engineering, Trinity Biomedical Science Institute, Trinity College Dublin, University of Dublin, Dublin, Ireland

 MA, 0000-0002-4540-4434

Many cell types are known to modulate their behaviour in response to changes in material stiffness; however, little is known about how stiffness affects corneal epithelial cells. This study aims to investigate the response of a corneal epithelial cell line to polydimethylsiloxane (PDMS) substrates with a range of Young's moduli from 10 to 1500 kPa. Cellular morphology, proliferation, differentiation and mechanobiology were examined. Cells grown on PDMS adopted the typical cobblestone morphology exhibited by the corneal epithelium. Proliferative markers pERK and Ki67 were higher in cells cultured on stiffer substrates compared with those on softer substrates. Material stiffness was also found to influence the cell phenotype with cells on stiffer substrates having higher cytokeratin 3 gene expression, a mature epithelial marker, while cells on softer substrates expressed more cytokeratin 14, a basal epithelial marker. Cells grown on softer substrates also displayed higher levels of focal adhesions and intermediate filaments compared with cells on stiff substrates. This research will aid in designing novel biomaterials for the culture and transplantation of corneal epithelial cells.

## 1. Introduction

Damage to the corneal epithelium can occur due to a variety of conditions including limbal stem cell deficiencies or physical abrasion. Limbal-derived stem cells allow the epithelium to undergo renewal following injury by migrating into the damaged region and differentiating into epithelial cells. However, if the injury is too severe or if there is a lack of healthy stem cells in the limbus, then additional therapeutic approaches may be required to repair the ocular surface such as cell transplantation [1]. Allogeneic limbal tissue may be taken from a donor and

applied directly to a patient to provide a new stem cell source [2]; however, a lack of suitable donor tissue can limit this option. The alternative is to isolate and culture limbal epithelial stem cells *in vitro* and then to transplant these cells on a biomaterial carrier. This approach has the advantages of allowing a higher number of cells to be transplanted and allowing autologous cells from a patient biopsy to be used. However, optimization of the culture environment, including the physical substrate onto which the cells are adhered, is required to control the cell phenotype.

When culturing cells on a substrate or fabricating biomaterials for cell transplantation, it is important to consider the mechanical characteristics of the materials since these will influence how the cells behave [3]. Examples of how material stiffness affects cells include by directing the differentiation of mesenchymal and adipose stem cells [4,5], influencing the proliferation, migration and resistance to chemotherapy of cancer cells [6,7] and modulating inflammatory cells such as macrophages [8]. In the cornea, only a small number of studies have examined the role that material stiffness has on the behaviour of corneal epithelial and limbal cells [9]. Factors affecting epithelial cells that have been examined in response to changes in stiffness include cell migration and viability [10] as well as stratification and differentiation [11], generation of tractional force by cells [12], nuclear yes-associated protein (YAP) expression [13] and cytokeratin expression [14]. One limiting factor with these studies is that since they use either polyacrylamide or collagen gels as substrates, only a narrow range of stiffness values could be examined. The mechanical environment of corneal epithelial cells can vary with the cells in contact with soft substrates such as the basement membrane (modulus $\approx$ 7.5 kPa) [15,16], stiffer substrates such as the corneal stroma (0.17–1.5 MPa) [5,17–19] following the loss of Bowman's layer after laser photorefractive keratectomy [20] or even stiffer substrates such as an amniotic membrane (approx. 2.6 MPa) [21].

The aim of this study was to examine the influence of material stiffness on a limbal-derived epithelial cell line using a wide range of stiffness values at days 3 and 7. The corneal epithelium is replaced after approximately 7 days; therefore, an early and late-stage response to stiffness was studied to determine how cells responded at different stages in their typical life cycle *in vivo* [22]. Polydimethylsiloxane (PDMS) was used to fabricate substrates with Young's modulus ranging from 10 to 1500 kPa. No protein coating was used for this study so as to eliminate the influence of the coating on the cellular phenotype. Cell morphology, differentiation, proliferation and mechanobiological responses were assessed to determine the relationship between cell behaviour and material stiffness. Cells cultured on tissue culture plastic (TCP) were used as the control group for this study.

# 2. Material and methods

## 2.1. PDMS fabrication

PDMS blends of varying stiffness were made using a commercially available product of Sylgard 184 and Sylgard 527 (Dow Corning). The softest blend of Sylgard 527 was prepared as per the manufacturer's instructions mixing equal quantities of parts A and B. Sylgard 184, the stiffest substrate, was also prepared as per the manufacturer's instructions blending 10 parts base to 1 part curing agent. Equal amounts of Sylgard 527 and Sylgard 184 were blended to create a 1 : 1 ratio of the stiffest and softest PDMS blends to make the medium group. A blend of five parts 527 to one part 184 was prepared and used as the medium-soft group. All samples were centrifuged at 650$g$ for 5 min to reduce air bubbles before casting into 6 or 24-well plates. Samples were cured at 60°C overnight. Dog-bone moulds were used to cast samples for tensile testing. The groups used in this study were a TCP control, stiff, medium, medium-soft and soft.

For the purposes of immunocytochemistry, PDMS groups were spin coated onto 12 mm glass coverslips to allow for confocal microscopy imaging. Each group was spin coated onto coverslips at 863$g$ for 15 s using a spin coater. The thickness of PDMS spin-coated samples was determined using white light interferometry. After spin coating, a scratch was made in each sample as an indirect measure of thickness to ensure that cells were sensing the substrate and not the glass.

## 2.2. Mechanical characterization

Young's modulus of each sample was determined using a Zwick tensile tester with each blend tested at least four times. A 10% strain was applied using a 5 N load cell at a loading rate of 4 mm min$^{-1}$. Young's modulus was calculated from the slope of the linear elastic region of the stress–strain curves.

## 2.3. Cell culture

A vial of hTCEpi (telomerase-immortalized human corneal epithelial cell line) was used for this study, developed from human limbal corneal epithelial cells (Evercyte). The cells ectopically express the catalytic subunit of human telomerase. The cells were seeded at 5000 cells cm$^{-2}$ onto PDMS-coated plates of various stiffness and cultured using Keratinocyte Growth Medium 2 fully supplemented (KGM-2, PromoCell) containing low calcium (0.06 mM) and no serum or antibiotics. Cells were cultured for 3–7 days and analysed using RT–PCR, immunocytochemistry and western blot.

## 2.4. Cell adhesion

Cells were imaged on a brightfield microscope, 4 h after seeding onto PDMS and TCP and counted using the cell counter on ImageJ. The number of adhered cells cm$^{-2}$ was calculated and used to determine the percentage of adhered cells using the initial seeding density value of 5000 cells cm$^{-2.}$

## 2.5. Contact angle

The contact angle of each PDMS substrate and TCP was determined using an FTA125 contact angle analyser (First Ten Angstroms, Inc.). The contact angle of each material was calculated via a static sessile drop technique using water. Each material was tested for contact angle three times with the contact angle determined after approximately 10 s of dropping water onto the surface to ensure the droplet was static. An average of the three measurements was used to determine the contact angle of each PDMS substrate and TCP.

## 2.6. Presto Blue

A PrestoBlue® assay (Thermo-Fisher Scientific) was used to assess metabolic activity. This kit was used as per the manufacturer's instructions. Media were aspirated from each well and a 1 : 10 mixture of PrestoBlue reagent and corneal epithelial media was prepared and added to wells. This was incubated at 37°C for 1 h, the reagent and media were placed in triplicate into a 96-well plate as well as a blank and absorbance read at 570 nm.

## 2.7. RT–PCR

RNA was isolated from monolayer cultures using Trizol (Invitrogen). One millilitre of Trizol was added per well (on a 6-well plate) followed by cell scraping. Cells were collected in RNAse-free Eppendorf tubes and snap frozen in liquid nitrogen. Samples were transferred to −80°C for storage until further use. Cells were placed on ice to allow them to thaw, 200 µl of chloroform was added to each tube and centrifuged at 12 000$g$ at 4°C. RNA located in the upper phase was transferred to an RNAse-free tube, isopropanol was added at the same volume as well as 4 µl Glycoblue to allow the visualization of the RNA, and samples were stored at −20°C overnight. The tubes were placed on ice and centrifuged at 12 000$g$ at 4°C for 15 min. A visible blue RNA pellet formed, supernatant was discarded and the tubes were dried. One millilitre of 70% ethanol was added to wash the pellet. Another centrifugation step was performed, ethanol was removed and the pellet air dried. RNAse-free water (11 µl) was used to dissolve the pellet. A NanoDrop-1000 was used to determine RNA yield and purity.

Transcription of mRNA to cDNA was performed using a high-capacity cDNA reverse transcription kit (Invitrogen). The 5.8 µl of mastermix and 14.2 µl of diluted RNA (500 ng) were added to tubes and placed in a thermocycler, the reaction set-up was as follows: 10 min at 25°C, 2 h at 37°C, 5 min at 85°C, 1 min at 4°C. Tubes were stored at −20°C until qPCR step. Quantitative PCR was performed with TaqMan reagents, 4.5 µl cDNA, 5 µl TaqMan universal mastermix II and 0.5 µl primer. The following primers were used: CK3 (Hs00365080_m1), CK14 (Hs00265033_m1), ΔNP63 (custom-made primer adapted from [23]), ABCG2 (Hs01053790_m1) and GAPDH (Hs02758991_g1). All samples were run in triplicate with a GAPDH housekeeping gene control. Fold change expression was calculated using the ΔΔCt method with TCP as control.

## 2.8. Western blot

To evaluate the expression of pERK, TERK, cytokeratin 3 (CK3), cytokeratin 14 (CK14) and NP63 protein in the cells, western blot analysis was performed at day 7. Cell lysates were isolated from monolayer culture using RIPA Lysis buffer with a phosphatase inhibitor cocktail followed by cell scraping. Samples were centrifuged

at 15 870*g* for 5 min and the supernatant was used for western blotting. The cell lysates were subjected to sodium dodecyl sulfate–polyacrylamide gel electrophoresis (SDS–PAGE) using a precast 12% polyacrylamide gel at 200 V for 40 min in an electrophoresis rig (Bio-Rad). A molecular weight ladder was added to each gel, and proteins were transferred to polyvinylidene difluoride (PVDF) membranes via semi-dry transfer (Thermo-Fisher). Membranes were blocked in 3% bovine serum albumin (BSA) in Tris-buffered saline (TBS) and 1% Tween 20 (pH 7.6) for 1 h at room temperature. The membranes were incubated with phospho-p44/42 MAPK (137F5) rabbit monoclonal antibody (mAb) (ERK 1/2)(Th202/204) antibody #9101 and p44/42 MAPK (L34F12) mouse mAb (ERK 1/2) antibody #9102 at 1 : 1000 in 3% BSA for 12 h at 4°C to analyse expression of phosphorylated and total ERK protein, respectively. Other membranes were incubated with anti-cytokeratin 3 mouse mAb (ab77869—abcam) at 1 : 500 dilution in 3% BSA for 12 h at 4°C. Other membranes were incubated with anti-cytokeratin 14 (MAC3232—Sigma-Aldrich) at 1 : 1000 dilution and anti-NP63 (619 002—Biolegend) at 1 : 500 dilution in 3% BSA for 12 h at 4°C. The loading control for all proteins was GAPDH (ab9484—Abcam) at 1 : 1000 in 3% BSA. Membranes were washed three times for 5 min with TBS and 1% Tween 20 followed by secondary antibody incubation for 1 h at room temperature. All secondary antibody dilutions were double that of the primary. Anti-rabbit IgG, horseradish peroxidase (HP) linked antibody (Cell Signalling) was prepared at 1 : 1000 for NP63 and 1 : 2000 for ERK and GAPDH membranes, respectively, in TBS and 1% Tween 20. Rabbit anti-mouse IgG HP (ab6728—Abcam) was prepared at 1 : 1000 in TBS and 1% Tween 20 for CK3 detection. Membranes were washed three times for 5 min with TBS and 1% Tween 20 and the membranes developed using an immunodetection kit (enhanced chemiluminescence western blotting substrate—Thermo-Fisher) and developed using GelDoc system (Bio-Rad). Densitometry was performed using ImageJ software and graphed using GraphPad Prism 7.

## 2.9. Immunocytochemistry

Cells were fixed using 4% paraformaldehyde (PFA) for 15 min at room temperature followed by rinsing in phosphate-buffered saline (PBS) three times and stored in PBS until ready to stain. Cells were blocked and permeabilized using 2% fetal bovine serum (FBS) and 0.5% Triton-X in PBS (blocking buffer) for 2 h at room temperature, followed by primary antibody incubation at 4°C for 12 h at 1 : 10 blocking buffer in PBS (antibody buffer). The following antibodies and dilutions were used: anti-Ki67 (ab15580—Abcam) was made up at 1 : 1000 dilution, anti-ABCG2 (sc-58222—Santa Cruz) at 1 : 500 dilution, anti-NP63 (619 002—Biolegend) at 1 : 500 dilution, mouse anti-vinculin (ab18058—Abcam) at 1 : 1000 dilution, rabbit anti-vimentin (ab92547—abcam) at 1 : 1000 dilution and anti-pYAP (Ser127) antibody #4911—Brennan and Co at 1 : 100 dilution.

Cells were washed three times with antibody buffer to remove excess primary antibody and then incubated with a secondary antibody at double the primary antibody dilution for 2 h at room temperature. For ABCG2 and vinculin, goat anti-mouse IgG H&L (Alexa Fluor® 488) (ab150113—Abcam) was used. For all other proteins, donkey anti-rabbit IgG H&L (Alexa Fluor® 488) (ab150073—Abcam) was used. Cells were also stained for actin to visualize the cytoskeleton using Phalloidin-TRITC (Sigma-Aldrich) at 1 : 1000 in parallel to the secondary antibody incubation. All groups were also stained for cellular nuclei using fluroshield with DAPI (Sigma-Aldrich). Cells were imaged using a confocal microscope. Day 3 cells were imaged at a higher magnification to appreciate cellular morphology and distribution of proteins tested. At day 7, a lower magnification was used to appreciate the confluent monolayer formed. The mean fluorescence intensity was determined using ImageJ software and corrected for background.

## 2.10. Statistical analysis

All experiments were carried out in triplicate, statistical analysis and outlier calculation was performed using GraphPad Prism software. Data are presented as mean ± standard deviation (s.d.), significance was calculated either via one-way or two-way ANOVA with the post-Tukey test, significance deemed as $p \leq 0.05$ for all datasets.

# 3. Results

## 3.1. Material characterization

The thickness of all samples that were spin coated onto glass coverslips was between 10 and 25 µm with no significant differences between groups (figure 1*a*). Each PDMS group was tensile tested and the elastic modulus obtained using the slope of the linear regression on the stress–strain curve. The stiff group had

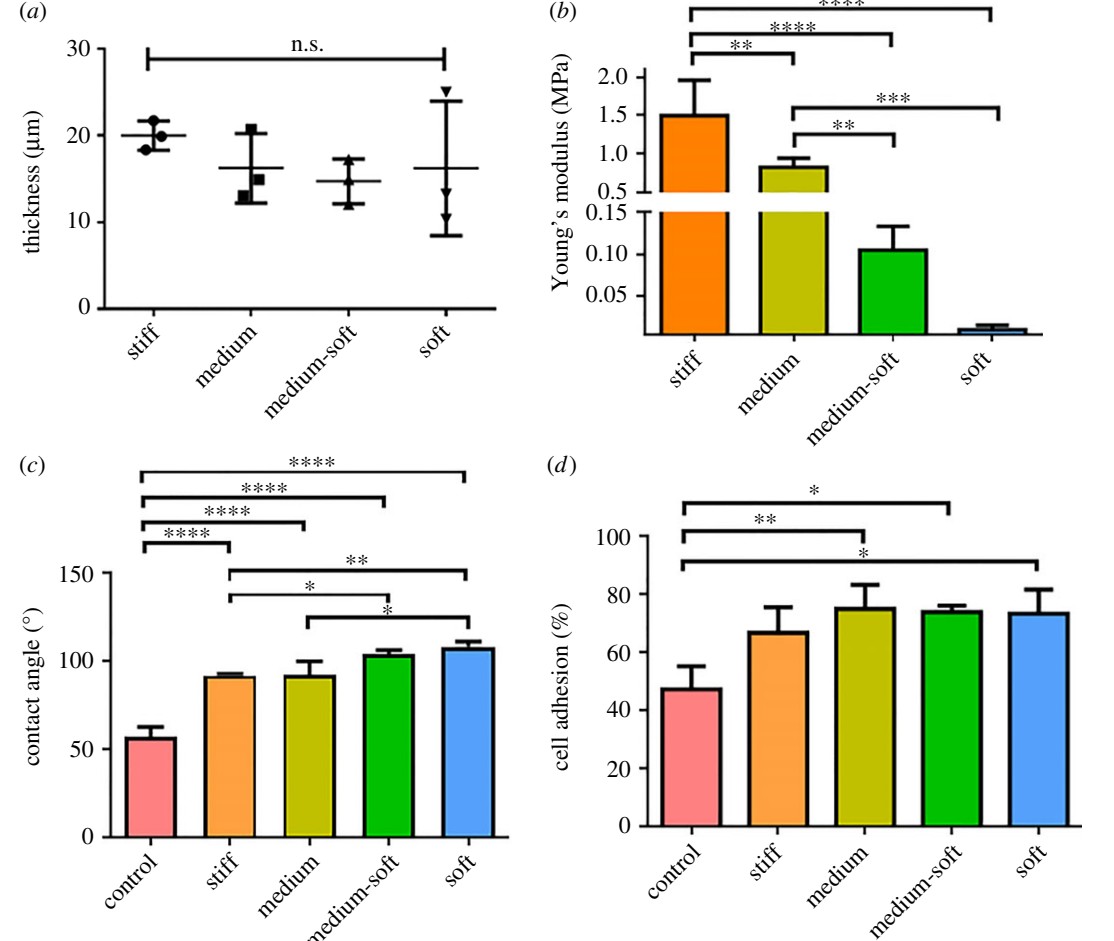

**Figure 1.** Characterization of PDMS substrates. (*a*) Thickness of spin-coated PDMS; (*b*) Young's modulus of PDMS; (*c*) contact angle of PDMS and TCP; (*d*) percentage of cell adhered to substrates after 4 h seeding. Data are presented as mean + s.d., significance calculated via one-way ANOVA with post-Tukey test, $N = 4–6$, $*p \leq 0.05$, $**p \leq 0.01$, $***p \leq 0.001$, $****p \leq 0.0001$.

an average elastic modulus of 1500 kPa, the medium group 820 kPa, the medium-soft group 105 kPa and the soft group 10 kPa. The stiff group had a significantly higher elastic modulus that all other groups. The medium group had a significantly higher elastic modulus than the medium-soft and soft group (figure 1*b*).

Contact angle was measured for all groups before cell seeding. The control group had a significantly lower contact angle than all the PDMS groups. The stiff group had a significantly lower contact angle than the medium-soft and soft group. The soft group had a significantly higher contact angle than the medium group (figure 1*c*).

The percentage of adhered cells was calculated 4 h after seeding. Between each PDMS group, no significant differences in adhesion were observed. The control group displayed significantly less adhesion compared with the medium, medium-soft and soft groups (figure 1*d*).

## 3.2. Cell proliferation and metabolic activity

After 7 days in culture, all cells on PDMS had a significantly higher rate of cell metabolic activity compared with cells on the control cell culture plastic (figure 2*a*). Between PDMS test groups, the stiff group had a significantly higher cell metabolic activity when compared with the soft group.

The proliferative marker ERK was analysed using western blot at day 7, the active form of this protein occurs post-transcriptionally; therefore, a western blot was performed to examine the phosphorylation of the protein and subsequent activation. Phosphorylated ERK (pERK) was normalized to total ERK (TERK) and a housekeeping protein GAPDH rather than β-actin as the expression of this protein was seen to be affected by the substrates (figure 2*b*). Densitometry analysis was performed to quantify differences in protein expression between the groups (figure 2*c*). The medium group showed the highest-level activation of pERK at day 7 and was significantly higher than the other groups. The least pERK expression was observed in the medium-soft and soft groups.

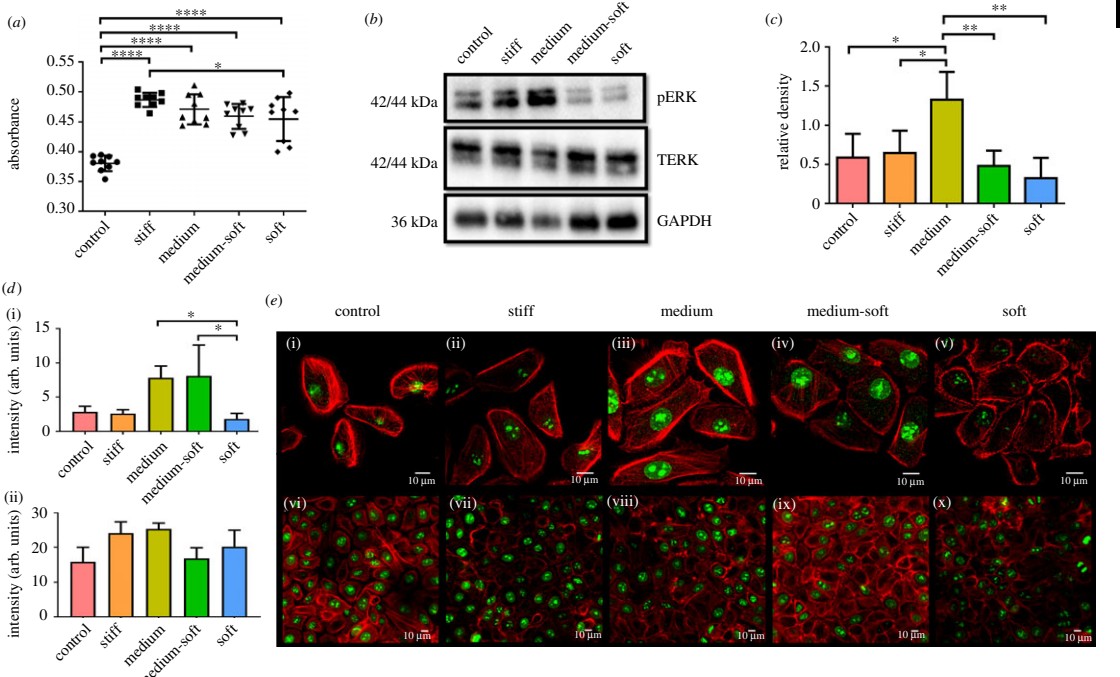

**Figure 2.** Metabolic activity and proliferation of cells in response to different substrate stiffness: (*a*) metabolic activity of cells after 7 days in culture; (*b*) western blot of cells grown on different substrates for pERK, TERK and GAPDH proteins at day 7; (*c*) densitometry analysis of western blot data at day 7, all data normalized to TERK and GAPDH; (*d*) fluorescence intensity of Ki67 staining at day 3 (i) and day 7 (ii); (*e*) Ki67 staining of cells grown on different substrates at day 3 (i)–(v) and day 7 (vi)–(x) (scale bar, 10 μm). Data are presented as the mean (±s.d.), significance calculated via one-way ANOVA with the post-Tukey test, $N = 3–6$, *$p \leq 0.05$, **$p \leq 0.01$, ****$p \leq 0.0001$.

Ki67, a marker of actively proliferating cells, was examined by immunocytochemical staining (figure 2*e*). At day 3, the control and stiff groups showed similar levels of Ki67 staining, indicating a similar proliferative rate. The medium group at day 3 showed the highest quantity of Ki67, significantly so compared with the soft group, indicating a more proliferative phenotype at this stiffness (figure 3 *d*(i)). The medium-soft displayed a significantly higher level of Ki67 compared with the soft group indicating decreased proliferation on soft substrates. At day 7, the control, stiff and medium groups showed similar levels of Ki67 staining. The medium-soft and soft group again had less Ki67 staining indicating less proliferative activity (figure 2*d*(ii)).

## 3.3. Cell differentiation

The mature corneal epithelial marker CK3 was examined using RT–PCR, western blot and immunocytochemistry. There was significantly higher gene expression of CK3 in the stiff group compared with the control, medium-soft and soft groups (figure 3*a*), indicating that stiffer substrates promote differentiation towards a more mature corneal epithelial phenotype. However, this was not detected on a protein level with both immunocytochemical staining and western blotting being negative for CK3.

Gene expression of CK14, a marker of hemidesmosome formation by basal epithelial cells, in the stiff, medium-soft and soft groups was significantly higher than the control group (figure 3*b*) at day 7. The medium-soft and soft groups also had significantly higher expression compared with the medium group. In western blot analysis (figure 4*a*), the medium-soft group expressed this protein significantly higher compared with all other groups. The soft group also expressed this protein higher than other groups, but this was not significant. These results suggest the medium-soft group produced a basal epithelial phenotype.

The stem cell marker ATP binding cassette subfamily G member 2 (ABCG2) was examined at days 3 and 7 using RT–PCR (figure 3*c*). No significant changes were seen in gene expression in any of the groups. However, at day 3, over all the medium-soft and soft group produced the highest level of ABCG2 gene expression but by day 7, the stiff and soft group had the highest level of ABCG2 gene expression albeit not significant. Immunocytochemical analysis was also carried out for this protein at day 7 (figure 5*a*–*e*). Nuclear localization was seen in all groups with the highest expression observed in the medium group (figure 5(i)). However, in general, all groups expressed this marker at a similar level.

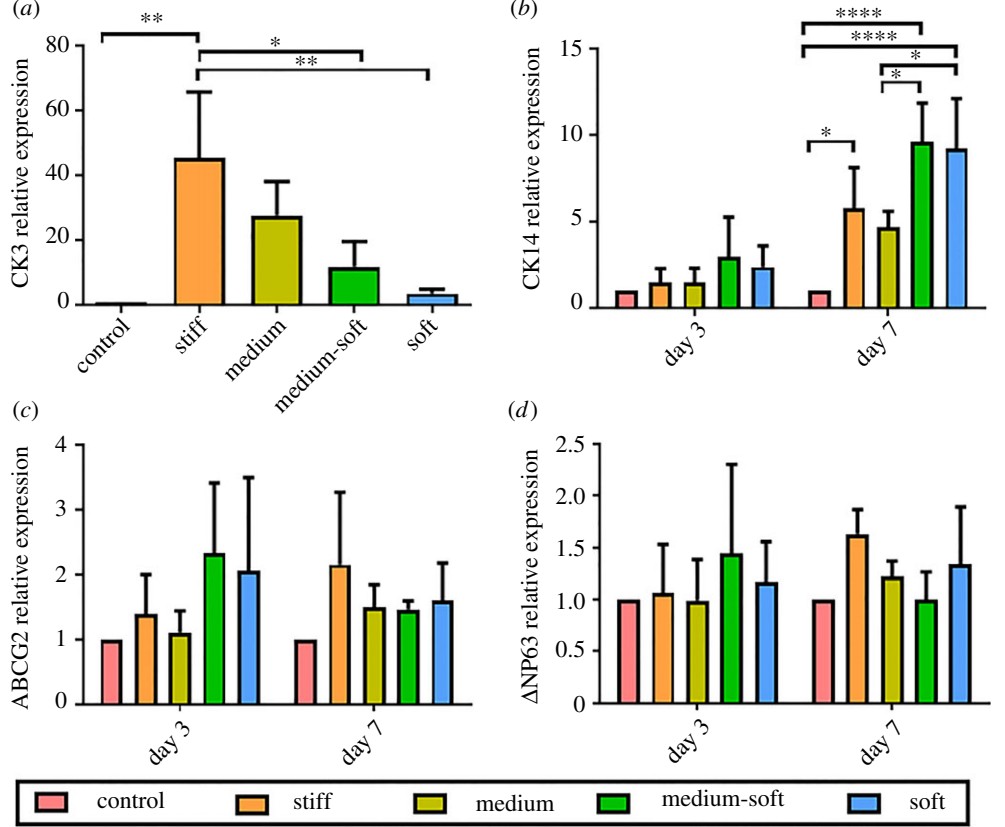

**Figure 3.** Real-time PCR to determine relative expression after 3 and 7 days: (*a*) CK3 gene expression at day 7; there was no detection of CK3 at day 3 in any groups; (*b*) CK14 gene expression; (*c*) ABCG2 gene expression; (*d*) $\Delta$NP63 gene expression. Data are presented as the mean ($\pm$s.d.), significance calculated via one-way ANOVA for CK3 and two-way ANOVA for CK14, $\Delta$NP63 and ABCG2 with the post-Tukey test, $N = 3$, *$p \leq 0.05$, **$p \leq 0.01$, ****$p \leq 0.0001$.

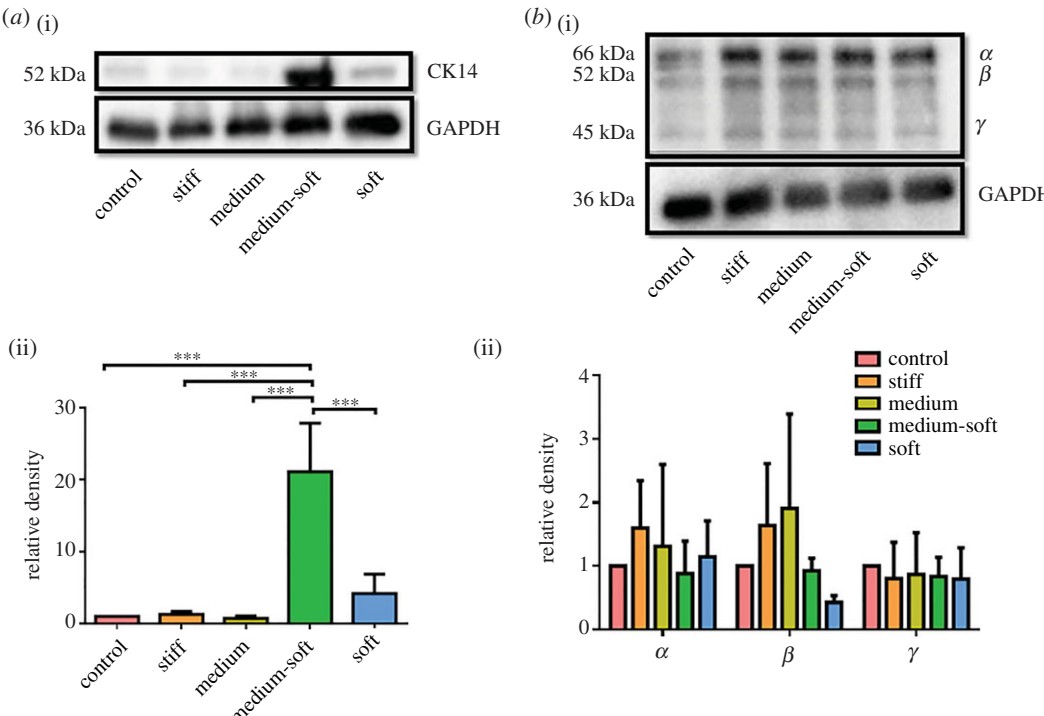

**Figure 4.** Western blot of CK14 and isoforms of $\Delta$NP63 at day 7: (*a*)(i) Sample western blot and (ii) densitometry analysis for CK14 and GAPDH; (*b*)(i) sample western blot and (ii) densitometry analysis of each NP63 isoforms. Data are presented as the mean ($\pm$s.d.), significance calculated via one-way ANOVA with the post-Tukey test, $N = 3$–4, ***$p \leq 0.001$.

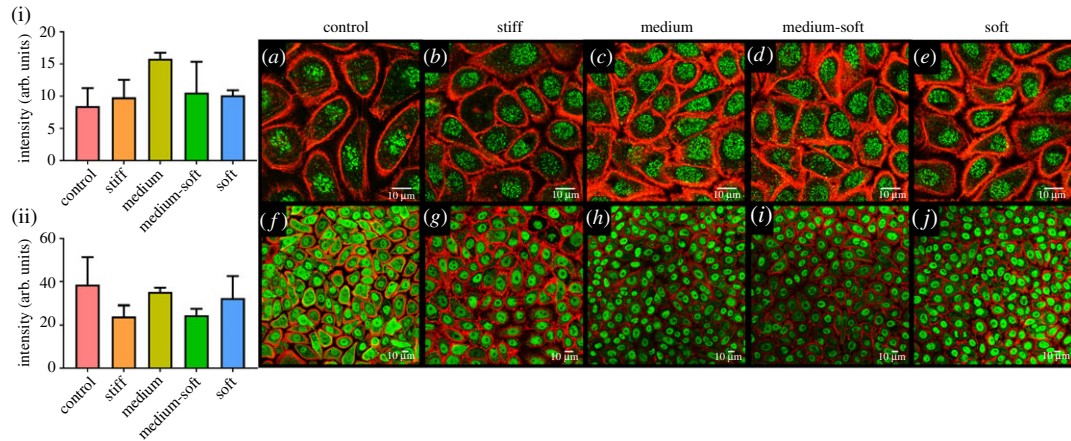

**Figure 5.** Immunocytochemical staining for stem cell markers after 7 days. (i) Fluorescence intensity of ABCG2 staining and (ii) NP63 staining. Top row (*a–e*) stained in green for ABCG2 protein; second row (*f–j*) stained in green for NP63. All cells were counterstained with f-actin (red) (scale bar, 10 μm). Data are presented as the mean (±s.d.), significance calculated via one-way ANOVA with the post-Tukey test, $N = 3$.

A second marker for a stem cell phenotype NP63 was examined. At day 7, gene expression analysis for ΔNP63 showed no significant difference between groups (figure 3*d*). However, a similar expression pattern to ABCG2 was observed with the medium-soft and soft group producing the highest level of ΔNP63 gene expression but by day 7, the stiff and soft group had the highest level of ΔNP63 gene expression albeit not significant.

Due to the presence of isoforms associated with different phenotypes in the corneal epithelium, a western blot was performed. Figure 4*b* shows the three isoforms of NP63 α, β, γ with the housekeeping protein GAPDH used as a loading control. Densitometry analysis showed that no significant differences were observed between groups. However, α and β isoforms were expressed at a higher level in the stiff and medium groups (figure 4*b*(ii)). Nuclear localization of this protein is an indication of its activation in cells. Therefore, immunocytochemical analysis was performed (figure 5*f–j*). The control and stiff group showed cytosolic and nuclear expression of NP63. The softer groups had little to no cytosolic expression with all cells showing a nuclear localization of the protein indicating a stem cell phenotype in softer substrates. The fluorescent intensity was highest in the TCP group indicating more staining on the cytoplasm (figure 5(ii)); however, no significant changes were observed between groups.

## 3.4. Mechanobiology

Mechanobiological responses of cells to stiffness were assessed at days 3 and 7 (figure 6). At day 3, cells grown on TCP (control group) displayed a smaller, more circular cell shape with little actin and fewer cellular projections compared with the cells grown on the different PDMS formulations. By day 7, these cells displayed projections and a polygonal shape but had more extracellular space compared with the other groups.

The stiff group displayed more of a spread cellular shape at day 3 (figure 6*b*) with visible filopodia extending from the cells edge and increased actin production compared with control. By day 7 (figure 6*g*), cells had little extracellular space with compacted cell morphology.

The medium group at day 3 (figure 6*c*) displayed stress fibres extending throughout the cell as well as longer cellular projections between each cell. A more spread cellular phenotype was also evident in this group with circular actin formations. Lamellipodia were evident in this group extending from the cells edge towards neighbouring cells. At day 7 (figure 6*h*), there were larger nuclei compared with the stiff and control group but less stress fibre formation across the cell compared with the cells at day 3.

The medium-soft group had a similar cell shape to the medium group with more pronounced stress fibres at day 3 (figure 6*d*). The cells also appear to be longer and more spread in comparison with all other groups. Cellular projections were not as evident in this group but rather cellular sheets extending from the cells were visible. By day 7, the medium-soft group (figure 6*i*) had more stress fibre formation compared with all other groups and increased nuclear size, similar to cells grown on medium stiffness. Decreased formation of circular structures in the actin cytoskeleton also occurred at day 7 in this group.

The soft group had less striking actin filaments with more of a triangular cell shape evident at day 3 (figure 6*e*). These cells displayed a sheet-like projection from the cells along with these projections which

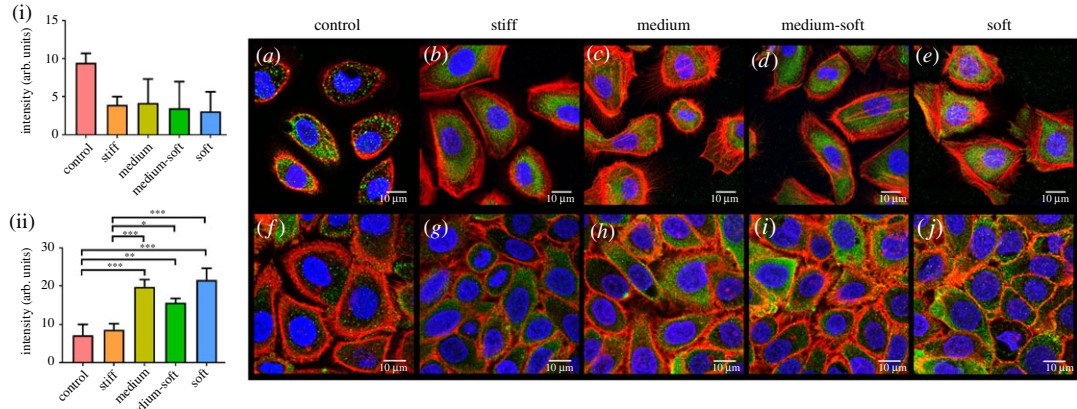

**Figure 6.** Immunocytochemical staining for vinculin. (i) Fluorescence intensity of vinculin at day 3 and (ii) day 7. (*a*–*e*) Cells stained in green for vinculin at day 3. (*f*–*j*) Cells at day 7. All cells were counterstained with f-actin (red) and DAPI (blue) (scale bar, 10 µm). Data are presented as mean (±s.d.), significance calculated via one-way ANOVA with post-Tukey test, $N = 3$, *$p \leq 0.05$, ** $p \leq 0.01$, ***$p \leq 0.001$.

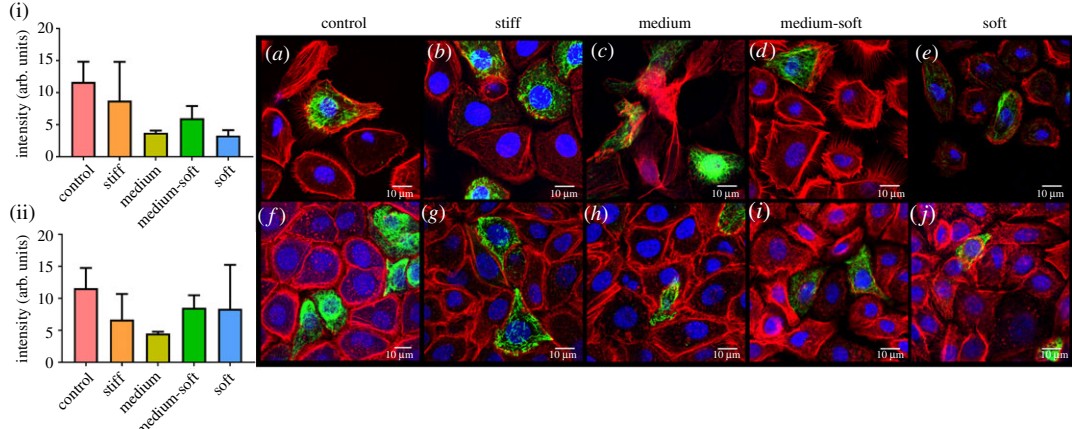

**Figure 7.** Immunocytochemical staining for vimentin. (i) Fluorescence intensity of vimentin at day 3 and (ii) day 7. (*a*–*e*) Cells stained in green for vimentin at day 3. (*f*–*j*) Cells at day 7. All cells were counterstained with f-actin (red) and DAPI (blue) (scale bar, 10 µm).

were seen in other groups, in particular the medium group; however, this was a lot more apparent in the soft group. The soft group at day 7 (figure 6*j*) had very little stress fibre formation and little to no circular actin formations. The cells appeared to have a much more defined cell membrane connection between each other and larger nuclei similar to that seen on the medium group at day 7.

All test groups showed a similar morphology at day 7 with more compacted cell phenotypes and less extracellular space compared with control group cells. Stiff, medium and medium-soft groups contained more actin between cells compared with the soft group and displayed circular structures in their actin cytoskeleton. The medium-soft group had the most stress fibre formation at day 7 compared with all other groups. The soft group had the least actin compared with all other groups as well as the least extracellular space.

Focal adhesions were examined by staining for vinculin (figure 6). At day 3, vinculin was high in the control group (figure 6(i)) and more visible at the cells edge (figure 6*a*). In the cells grown on PDMS, vinculin was more diffuse in the cytoplasm with the medium stiffness group showing the least focal adhesions (figure 6*a*–*e*); however, this was not significant (figure 6(i)). At day 7, the number of focal adhesions per cell decreased in the control group and was significantly higher in the medium, medium-soft and soft groups compared with the control (figure 6(ii)) with some cells displaying more focal adhesions at the cells edge. The stiff group had significantly less vinculin compared with the other PDMS groups at day 7.

Vimentin, an intermediate filament (IF) protein was examined at days 3 and 7 (figure 7). The control and stiff groups had the most vimentin, with IFs extending over the cellular nuclei in the medium group

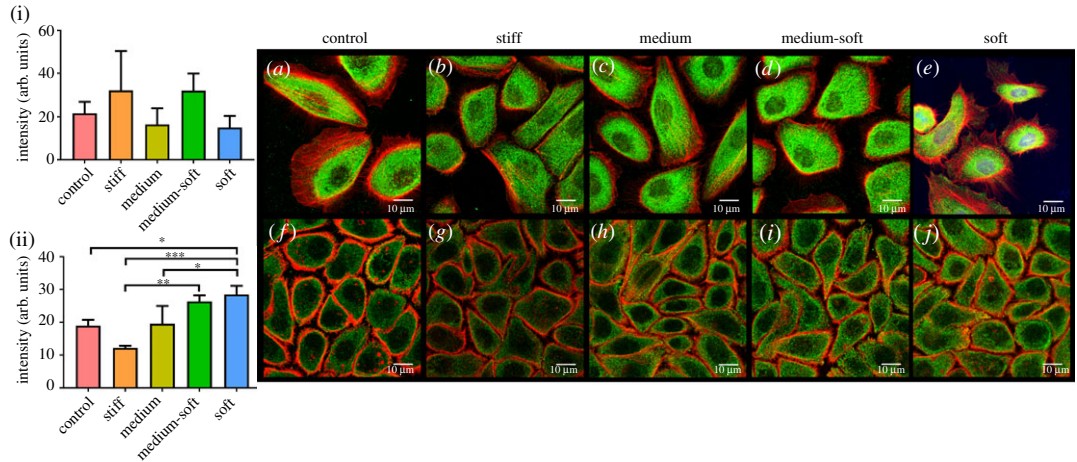

**Figure 8.** Immunocytochemical staining for phosphorylated YAP (pYAP) protein. (i) Fluorescence intensity of pYAP at day 3 and (ii) day 7. (a–e) Cells stained in green for pYAP at day 3. (f–j) Cells at day 7. All cells were counterstained with f-actin (red) (scale bar = 10 μm). Data are presented as the mean (±s.d.), significance calculated via one-way ANOVA with the post-Tukey test, $N = 3$, *$p \leq 0.05$, **$p \leq 0.01$, ***$p \leq 0.001$.

at day 3. No significant changes in vimentin expression were observed at either time points (figure 7(i),(ii)). The soft group had the lowest number vimentin-positive cells and the lowest production of the protein within the cell. At day 7, the control group cells displayed the highest level of IF expression as well as the highest number of vimentin-positive cells.

At days 3 and 7, the activated phosphorylated form of yes-associated protein (pYAP) was studied (figure 8). At day 7, the stiff group showed the lowest amount of pYAP expression with a significant decrease in expression compared with the medium-soft group. By day 7, the soft group had a significantly higher amount of pYAP production compared with control, stiff and medium groups (figure 8(ii)). Control and stiff groups had the lowest pYAP production at day 7. Nuclear localization of the protein was not evident in any groups. Images including nuclear staining using DAPI are shown in electronic supplementary material, figure S1.

# 4. Discussion

In this study, PDMS was used to evaluate the effect of substrate stiffness on the behaviour of corneal epithelial cells. In most previous PDMS stiffness studies, Sylgard 184 with differing ratios of base to curing agents was used to generate substrates with a narrow variation in stiffness. To achieve a wider range of stiffness values for this study, two PDMS solutions (Sylgard 184 and 527) were mixed at different ratios. This technique was originally developed by Palchesko et al. and has previously been used to examine the effect of substrate stiffness several different cell types [24,25] but not corneal epithelial cells. In this study, white light interferometry was used to determine the thickness of spin-coated substrates which was consistent across all substrates and did not exceed 10 μm. Previous studies have shown that single cells can only sense underlying materials through soft hydrogel materials at thicknesses below 10 μm which is measured by the degree of cell spreading [26]. Other studies have shown that cells can be influenced by underlying structures hundreds of micrometres away (up to 130 μm). However, this is influenced by the structure of the substrate in question with fibrous substrates that are similar to the fibrous nature of biological substrates allowing cells to propagate forces through the individual fibres over longer distances [27]. In this presented study, cells were spread at all stiffness and more rounded on the TCP group at day 3 (figure 6a–e), indicating that cells were sensing the PDMS substrate rather than the glass. This, coupled with the significant differences in gene and protein expression as well as the homogeneous non-fibrous nature of PDMS, indicated that the cells were reacting to the stiffness of the PDMS and not the underlying coverslip. Therefore, this study ensured that all spin-coated substrates exceeded this value.

To further characterize the PDMS substrates, water contact angle was used to measure hydrophobicity and how this influences adhesion of the cells. There was a significant difference between the hydrophobicity of TCP and PDMS. This result correlates with fewer cells adhering on the TCP compared with PDMS after 4 h despite TCP being less hydrophobic. While this suggests that hydrophobic surfaces may be better at supporting corneal epithelial cell adhesion, some studies have postulated that water

contact angle may not be an accurate predictor of biological responses to materials [28]. Values for the optimal contact angles for cell adhesion vary considerably between publications, most likely due to the use of different cell types and materials [29–32].

Most studies which examine cellular response to stiffness coat surfaces with biological substances such as collagen or laminin [4,33]; however, this study has shown that a more hydrophobic surface can still support corneal epithelial cell adhesion and that a coating may not be required to study cellular responses. Previous studies have shown that collagen coating can penetrate deeper on softer substrates compared with stiffer ones [34], which may affect cellular responses. This cell line is used for culturing on TCP, which is why this was chosen as the control group.

Stiffness was shown to influence the proliferation and cell metabolic activity of the cells. Only small variations in metabolic activity were detectable, probably due to the lack of sensitivity of the assay due to the colour changing quickly with time as well as its use as a measure of cell viability rather than cellular metabolism [35]. A clearer trend was detectable when proliferative markers such as Ki67 and pERK were examined. Expression of pERK was measured using western blot analysis at day 7 only, as higher protein yields were obtained at this point, which made it possible for the protein bands to be visualized compared with day 3. The stiff and medium groups had the highest level of Ki67, while the medium group had significantly higher pERK production compared with the other groups. The increase in proliferative markers in the stiffer groups suggests that cells on the softer substrates are slower cycling. In the eye, it is necessary for corneal epithelial cells to undergo proliferation to maintain homeostasis; however, a subpopulation of cells have been found to undergo a slower cycle that can accelerate if necessary to support homeostasis or response to damage [36]. Ki67 protein accumulation occurs in the S, G2 and M phase of the cell cycle and degraded in G1 and G0 (quiescence) of the cell cycle [37]. It is evident from Ki67 staining that although not actively proliferating or dividing, the number of cells present is similar among groups, which indicates cells in a quiescent state retaining their slow cycling yet proliferative abilities in repairing the ocular surface.

Material stiffness was shown to affect cell phenotype. *In vivo*, cells from the limbus proliferate and migrate along the basal region of the corneal epithelium cells where they undergo transient amplifying behaviour [38]. These cells then divide, producing mature, differentiated epithelial cells in the superficial region of the epithelium. To distinguish between different phenotypes, CK3 was used as a marker of a mature, differentiated phenotype and CK14 was used to identify transient amplifying cells like those found in the basal epithelium but are absent from the central cornea [14]. During wound healing, CK14-positive cells have been identified in the basal epithelium of mice [39]. Gene expression analysis at day 7 showed there was significantly more CK3 in cells on stiffer substrates than softer while expression of CK14 was higher on the softer substrates compared with the stiff. This indicates that stiffer substrates produce a more mature corneal epithelial phenotype while the softer substrates support a more stem-like phenotype. Previous research looking at substrate stiffness using collagen hydrogels yielded similar results with stiffer substrates displaying increased CK3 expression in bovine limbal stem cells, while softer substrates increased the expression of CK14. It is worth noting that the stiff group in that study was only 2.9 kPa compared with 1500 kPa in this presented study [14]. However, a recent study showed that using compressed collagen gels *in vitro*, a more mature phenotype expressing CK3 was observed in the stiffer group of $4.8 \pm 3.5$ MPa and a stem cell phenotype expressing CK15 in the softer group of $0.7 \pm 0.4$ MPa. This study used a laminin coating that may have influenced cellular response to stiffness [33].

Expression of markers associated with a limbal stem cell phenotype, ABCG2 and NP63, was also examined with no significant changes observed. Western blot analysis of NP63 isoforms associated with different phenotypes in the corneal epithelium also had no significant difference. However, using immunocytochemistry, it was clear that in softer substrates, nuclear localization of NP63 was higher than in stiff substrates. It has been shown that nuclear localization of NP63 is lost in differentiated superficial cells of the cornea [23], further supporting the hypothesis that softer substrates induce more of a limbal stem cell phenotype. It may be possible that differentiated corneal epithelial cells can still express genes for stem cell markers and allow the ocular surface to repopulate the stem cell niche if required. A recent study showed in a mouse model that differentiated committed cells in the central cornea have the ability to migrate back towards the limbus when this stem cell niche is removed and repopulate this niche [40]. Additionally, other papers have proposed a corneal epithelial stem cell hypothesis where stem cells that are distributed in the basal layer have stem cell functionality in regenerating the ocular surface, and these cells do not necessarily originate from the limbus [41]. Furthermore, a recent study showed using an *in vivo* model of wounded corneas that by softening these damaged corneas using collagenase a limbal epithelial stem cell-like phenotype can be promoted improving wound healing [33].

Immunocytochemistry was used to visualize the localization of pYAP in the cells. YAP is involved in the Hippo signalling pathway and in the cytoskeletal responses to substrate rigidity and topography along with its co-activator PDZ binding motif (TAZ) and has previously been investigated in the corneal epithelium [42]. YAP-mediated gene expression can lead to stem cell proliferation, apoptosis evasion and cell proliferation; however, YAP regulation by mechanical stress is not completely understood [43]. Regulation of YAP occurs through phosphorylation whereby unphosphorylated YAP is sequestered in the nucleus and acts as a transcriptional co-activator, phosphorylated YAP by a Lats kinase inhibits its functionality and sequesters YAP in the cytoplasm where it is targeted for subsequent degradation [44]. All groups displayed cytoplasmic distribution of pYAP with little nuclear staining. This was more pronounced at day 7, whereby all groups displayed a lower level of pYAP expression compared with day 3. This indicates that the degradation of pYAP occurs over 7 days, and perhaps earlier time points would be more beneficial to see when it is sequestered into the nucleus to act as a transcriptional co-activator. Immunocytochemical staining at day 7 also showed a significant increase in pYAP in the softer groups compared with the control and stiff group. This may indicate that softer substrates retain stem cell characteristics over a longer time compared with stiffer substrates. Ectopic expression of YAP has been shown to maintain stem cell phenotypes even under differentiation conditions [45].

Focal adhesions between cells and their surrounding matrix play an important role in dictating cell migration, cytoskeletal structure and cell signalling. In this study, vinculin was used to study localization of focal adhesions. At day 3, the control group showed the highest vinculin intensity, while all other groups had similar levels. Staining was localized to the cytoplasm rather than along the cells edge as would be expected for this. By day 7, this trend seemed to differ with the medium, medium-soft and soft groups producing significantly higher levels of vinculin compared with the control and stiff group. This finding suggests that at earlier time points, cells on softer substrates, which have also shown a stem cell phenotype, have a lower migratory capacity. However, by day 7, the production of focal adhesions is higher in these groups, suggesting that these cells migrate at a later stage, presumably to repopulate cells that have been terminally differentiated and shed from the ocular surface. However, as this study was performed in a monolayer, a stratified model would be required to confirm this finding.

Vimentin was examined, as it has been described as a highly motile early differentiating corneal epithelial cell marker associated with other early differentiating markers including ΔNP63α and α6 integrin in one study [46], and previous studies have shown it to be involved in wound healing of the corneal epithelium [47]. At day 3, more cells in the stiffer groups appeared to display this protein compared with the softer group, possibly indicating that cells on stiffer substrates have initiated differentiation towards a more mature epithelial phenotype. This would then correlate with gene expression analysis of CK3 at day 7.

The results of this study have implications for the design and application of biomaterials for culturing or transplanting limbal-derived epithelial stem cells. For example, amniotic membrane has been widely used for transplantation and *ex vivo* expansion of limbal stem cells; however, variations in the membranes' stiffness can result from donor variations and whether the pregnancy reached term [21], which could affect how the cells in contact with this material behave. One study that has investigated the effect of amniotic membrane stiffness on limbal cell behaviour concluded that stiffer substrates drive a more mature differentiated corneal epithelial phenotype, which would appear to agree with our findings [48]. Validation of these results using a primary cell source would be beneficial; however, the use of a coating for these studies also limits their findings, and the cell line used in this study has been validated as a model for corneal epithelial studies when compared with primary cells [23].

Overall, this study shows that over a wide range of stiffnesses reported in the literature, differentiation, proliferation, focal adhesion and IF expression is significantly affected. Future studies looking at cell surface receptors, on how these mechanical signals are relayed as well as earlier time points to determine pYAP response would further aid in our understanding of the corneal epithelial response to stiffness.

# 5. Conclusion

This study demonstrates that stiffness plays a major role on the differentiation, proliferation and morphology of limbal-derived epithelial cells using a corneal epithelial cell line as a model. Culturing cells on a material with Young's modulus in the range of 10–105 kPa would appear to be the most suitable for retaining the cells' stem-like characteristics. Limitations with this study include the use of

a cell line rather than primary cells, there was no air–liquid interface when culturing the cells and the topography of the PDMS substrates will differ from the cornea's basement membrane. Despite this, these findings could be applied when optimizing the design of biomaterials for limbal epithelial cell culture and transplantation.

Data accessibility. Data for this manuscript are available within the Dryad Digital Repository: https://doi.org/10.5061/dryad.ht76hdr9z [49].

Authors' contributions. S.M. carried out the laboratory work, analysed data, participated in study design and drafted the manuscript; M.A. conceived study, participated in study design and helped draft the manuscript. All authors gave final approval for publication and agree to be held accountable for the work performed therein.

Competing interests. The authors have no competing interest to declare.

Funding. The research is supported by funding from the European Research Council (ERC) under the European Union's Horizon 2020 research and innovation programme (grant agreement no. 637460) and from Science Foundation Ireland (grant no. 15/ERC/3269).

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
