## [Reviewer comments · Royal Society Open Science]

Review History

Decision letter (RSOS-191796)

04-Nov-2019

Dear Dr Ahearne:

It is a pleasure to accept your manuscript entitled "Influence of polydimethylsiloxane substrate stiffness on corneal epithelial cells" in its current form for publication in Royal Society Open Science. The comments of the reviewer(s) who reviewed your manuscript are included at the foot of this letter.

Kind regards,
Andrew Dunn
Senior Publishing Editor
Royal Society Open Science Editorial Office
Royal Society Open Science
openscience@royalsociety.org

on behalf of Professor Guy Genin (Associate Editor) and Dr Pietro Cicuta (Subject Editor).

Associate Editor Professor Guy Genin Comments to Author:

Associate Editor

Comments to the Author:

I enjoyed this well-written article about a very solid study and believe that the authors have done an excellent job of addressing previous critiques. My belief is that the article as it stands exceeds all standards for publication.

If the authors do have endurance for another round of suggestions, I have the following thought. Note that I do not consider this to be a requirement for my recommendation of publication. The authors have a very nice range of substrate compliances. This reviewer spent some time plotting out their data as a function of stiffness instead of on bar charts, and thought that the trends look really good. If the authors are inclined to do so, why not add in some graphs with modulus on the abscissa?

Great paper!
